# Active Nanointerfaces Based on Enzyme Carbonic Anhydrase and Metal–Organic Framework for Carbon Dioxide Reduction

**DOI:** 10.3390/nano11041008

**Published:** 2021-04-15

**Authors:** Qian Liu, Xinwei Bai, Huy Pham, Jianli Hu, Cerasela Zoica Dinu

**Affiliations:** Department of Chemical and Biomedical Engineering, West Virginia University, Morgantown, WV 26506, USA; ql0009@mix.wvu.edu (Q.L.); xb0001@mix.wvu.edu (X.B.); hgpham@mix.wvu.edu (H.P.); john.hu@mail.wvu.edu (J.H.)

**Keywords:** metal–organic frameworks, MOFs, enzyme, interface, active system, carbon dioxide

## Abstract

Carbonic anhydrases are enzymes capable of transforming carbon dioxide into bicarbonate to maintain functionality of biological systems. Synthetic isolation and implementation of carbonic anhydrases into membrane have recently raised hopes for emerging and efficient strategies that could reduce greenhouse emission and the footprint of anthropogenic activities. However, implementation of such enzymes is currently challenged by the resulting membrane’s wetting capability, overall membrane performance for gas sensing, adsorption and transformation, and by the low solubility of carbon dioxide in water, the required medium for enzyme functionality. We developed the next generation of enzyme-based interfaces capable to efficiently adsorb and reduce carbon dioxide at room temperature. For this, we integrated carbonic anhydrase with a hydrophilic, user-synthesized metal–organic framework; we showed how the framework’s porosity and controlled morphology contribute to viable enzyme binding to create functional surfaces for the adsorption and reduction of carbon dioxide. Our analysis based on electron and atomic microscopy, infrared spectroscopy, and colorimetric assays demonstrated the functionality of such interfaces, while Brunauer–Emmett–Teller analysis and gas chromatography analysis allowed additional evaluation of the efficiency of carbon dioxide adsorption and reduction. Our study is expected to impact the design and development of active interfaces based on enzymes to be used as green approaches for carbon dioxide transformation and mitigation of global anthropogenic activities.

## 1. Introduction

Self-sustainable carbon dioxide (CO_2_) atmospheric removal strategies that have high efficiency and feasibility for large-scale implementation as well as good environmental compatibility [1,2] are needed to circumvent ecological [3,4] and human [5,6] effects and to reduce or prevent dangerous anthropogenic interferences [7]. Current technologies for CO_2_ mitigation rely on postcombustion and CO_2_ sequestration, or amine-based scrubbing [8,9,10,11]. However, the drawbacks of such technologies are the high energy requirements needed for CO_2_ desorption, reduced regeneration of the adsorbent [3,4] or, for amine-based strategies, the loss of amine functionality [12,13]. Attempts to use alternative strategies based on ionic liquids [14,15,16], piperazine [17,18], and organic solvents [19,20] were considered; however, the high corrosiveness [21,22], volatility [18,23], and parasitic energy consumption of such processes [1,24] precluded their large-scale implementation. 

Carbonic anhydrases (CAs; EC 4.2.1.1) are a family of metalloenzymes known to catalyze the reversible conversion of CO_2_ into bicarbonate in the human body, all with ultrahigh turnover of up to 10^5^ s^−1^ [25]. Isolated CAs used in synthetic assays were shown to allow evaluation of a range of disorders from edema, to glaucoma, cancer, and osteoporosis [26,27,28,29], just to name a few. CAs were also proved to play critical roles in the proliferation, survival, and differentiation of pathogens such as protozoa, fungi, and bacteria [30,31], with differences in the infected host relative to the uninfected habitat being evaluated for the potential development of novel therapeutic agents [32]. Furthermore, due to their involvement in reversible hydration/dehydration and pH homeostasis gluconeogenesis, lipogenesis, and ureagenesis [33,34], respectively, CAs have been considered as candidates for biosensors applications [35,36,37,38]. Moreover, their capability to enhance CO_2_ capture and sequestration has been recently explored for the development of industrial applications based on membranes [39,40,41] to be used for contactors [39,40] and for supports of ionic liquid membranes [41]. Current active technologies rely on gas adsorption membrane (GAM) units [39,40] where the gas and liquid phases are spatially separated by a hydrophobic entity, with CA being immobilized on the liquid phase side. In such systems, the wetting relationship between the membrane and liquid solvent has proved critical for defining the overall membrane’s performance for CO_2_ adsorption [42,43]. However, low CO_2_ solubility in water, the active solvent maintaining enzyme’s functionality, continues to lead to intricate restrictions when large scale implementation of such CA-based systems [40,44] is considered. Moreover, challenges exist when aiming to identify any mass transfer limitations at its interface to thus relate the overall membrane’s transformation ability to its scale-up and operational stability [45].

We hypothesized that nanoporous interfaces with hydrolytic stability could not only allow for user-controlled CA immobilization but could also ensure enzyme’s activity and its stability retention for subsequent CO_2_ adsorption, all at ambient conditions. To test our hypothesis, we synthesized and used metal–organic frameworks (MOFs), a type of porous crystal materials constructed by linking metal node (metal ions or clusters) and organic linkers together through bond coordination, as scaffolds for both enzyme immobilization and interfaces functionality demonstration. Studies in our group and others have previously showed that such frameworks are suitable for enzyme surface immobilization because of their customizable synthesis [46], large surface area which allows for increased enzyme loading [47,48], and tunable surface features which permit hydrophilic or hydrophobic residues implementation [48,49,50]. We and others have also showed that resulting MOF–enzyme conjugates [46] preserved enzymes’ functionality to a larger extent than conjugates formed upon enzymes immobilized on other materials such as polyurethane foam [51], magnetic [52] and silica-based materials [53], or zeolites [54], respectively. Our proof-of-principle demonstration integrated in a lab-built bioreactor provides a functional interface for efficient CA functionality at ambient pressures and temperatures while allowing the study of kinetics efficiency, as well as system’s operational stability in user-set-up conditions. Lastly, our demonstration provides a customizable interface for large-scale implementation with functionality determined by both the enzyme and MOF distribution, respectively. 

## 2. Materials and Methods

### 2.1. Materials

The compounds 2,5-furandicarboxylic acid (FDCA, 98%, Thermo Fisher Scientific Chemicals, Inc., Ward Hill, MA, USA), aluminum chloride hexahydrate (99%, Acros Organics, Fair Lawn, NJ, USA), and sodium hydroxide (extra pure, Across Organics, Fair Lawn, NJ, USA) were used for metal–organic frameworks (MOFs) (i.e., MIL-160) synthesis. Aluminum oxide (Al_2_O_3_) filters with diameters of 13 mm and pore size of 20 nm were purchased from Separation Processes, Inc., Carlsbad, CA, USA and used as supports for membrane formation. Tris(hydroxymethyl) aminomethane (99%, Alfa Aesar, Fair Lawn, NJ, USA) was used for buffer preparation (Tris buffer, pH 7.5, 50 mM). Carbonic anhydrase (CA) from bovine erythrocytes (E.C.4.2.1.1) was purchased from Sigma-Aldrich Inc., St. Louis, MO, USA. Para-nitrophenol acetate (p-NPA, Across Organics, Fair Lawn, NJ, USA) was used as substrate for enzyme activity assessment. Bicinchoninic acid assay (BCA) kit (Thermo Fisher Scientific Chemicals, Inc., Ward Hill, MA, USA) was used to evaluate enzyme loading onto the surfaces. Acetonitrile (Thermo Scientific Chemicals, Inc., Ward Hill, MA, USA) was adopted as a solvent for p-NPA, while a mixture of nitrogen and carbon dioxide (Airgas, 95% N_2_ and 5% CO_2_, Morgantown, WV, USA) gases was used to evaluate the CO_2_ adsorption performance at the synthesized membrane interface. All chemicals were commercially available and used without further treatment.

### 2.2. Preparation of MIL-160/Al_2_O_3_ Hybrids 

FDCA-modified Al_2_O_3_ filters were used for synthesis of MIL-160/Al_2_O_3_ hybrids. For this, 10 mL of 5 mM FDCA solution in deionized water was first dosed onto the Al_2_O_3_ filters contained in a Teflon-lined stainless steel autoclave and subsequently heated at 50 °C for 24 h. Secondly, such modified filters were subsequently dosed in a Teflon-lined stainless steel autoclave in a solution of FDCA, aluminum chloride hexahydrate, and sodium hydroxide dissolved in deionized water (final concentrations of 55, 55, and 100 mM, respectively) at 100 °C for 12 h. After cooling to room temperature, the functionalized filter was taken out, washed extensively with deionized water to remove any of the unreacted species, dried overnight and at room temperature, and stored for further usage. 

### 2.3. Characterization of MOFs and Resulting Hybrids

The morphologies of the Al_2_O_3_, FDCA functionalized Al_2_O_3_ (FDCA/Al_2_O_3_) filters, and MIL-160/Al_2_O_3_ hybrids were investigated by field emission scanning electron microscopy (FESEM) performed on a S-4700 (Hitachi, Santa Clara, CA) with a cold field emission gun operating at 4 kV and 10 JA. Energy-dispersive X-ray spectroscopy (EDS) mapping was conducted for evaluating the elemental composition of the samples. Phase purity and crystallinity of the synthesized MOFs were confirmed by X-ray diffraction (XRD), with powders of MOFs being analyzed using a Bruker D8 ADVANCE X-ray diffractometer (Westborough, MA, USA) with CuKa radiation operating at 40 kV and 40 mA. XRD patterns were recorded in reflection mode, at room temperature, and under ambient conditions. Chemical composition of the Al_2_O_3_ and functionalized Al_2_O_3_ filters were evaluated in dried conditions by Fourier-transform infrared spectroscopy (FTIR) on a Digilab FTS 7000 (Digilab, Inc., Hopkinton, MA, USA) equipped with a diamond attenuated total reflection (ATR). Scans were collected in the range of 400–4000 cm^−1^ at a resolution of 4 cm^−1^; a total of 128 scans were co-added to form the final spectrum of each sample investigated. Experiments were performed in triplicates. The surface areas of Al_2_O_3_ filter and MIL-160/Al_2_O_3_ hybrids were determined from N_2_ adsorption analysis conducted on a Micromeritics Instrument sorption analyzer (ASAP 2020, Micromeritics Instrument Corporation, Norcross, GA, USA) using liquid nitrogen at 77 K. An atomic force microscope (AFM, Asylum Research, Goleta, CA, USA) and Si tips (AC240TS, 50 kHz, Asylum Research, Goleta, CA, USA) were used to investigate the morphology of the samples in AC mode. Scans of 20 µm × 20 µm area or lower were acquired in air; order 3 image flattening was applied. Height evaluation were performed using the Asylum software (Asylum Research, Goleta, CA, USA), with subsequent MATLAB analysis. 

### 2.4. CA/MIL-160/Al_2_O_3_ Membrane Preparation

The CA/MIL-160/Al_2_O_3_ membrane was prepared by loading CA onto the MIL-160/Al_2_O_3_ hybrids. For this, the hybrids were immersed in a 2 mL solution containing different enzyme concentrations (1.0, 0.5, and 0.1 mg/mL, respectively) in Tris buffer (50 mM and pH 7.5) and in a covered glass vials (Fisher Scientific, Ward Hill, MA, USA), with the mixture being shaken at room temperature and 100 rpm. After 2 h, the resulting (now) membrane was removed and used for the activity assay and CO_2_-related tests, respectively. Control experiments have also been performed using Al_2_O_3_ filters and incubating them with the above known representative concentrations of enzyme. 

### 2.5. Enzyme Loading Assessment

Standard BCA assay was performed to evaluate enzyme loading or the amount of enzyme loaded per the membrane. Briefly, 1 mL of the working reagent containing 50 parts of reagent A with 1 part of reagent B (reagents were provided stock with the BCA assay kit) was mixed with 50 μL of the remaining solution, i.e., solution used for the membrane or filter and containing the different enzyme concentrations (see above), and subsequently incubated at 37 °C for 30 min. Absorbance at 562 nm was recorded for each sample using an UV−VIS spectrophotometer (UV2600 Shimadzu) and compared to a calibration curve of known concentrations of the CA suspended in the working reagent. Every experiment was repeated at least 3 times. Loadings were calculated using Equation (1):Enzyme loading (mg/mg) = C_0_V_0_ − C_1_V_1_(1)
where C_0_ and V_0_ were the initial concentration and volume of CA solution, and C_1_ and V_1_ were the concentration and volume of CA solution after the immobilization process, respectively.

### 2.6. Enzyme Activity at the Designed Interfaces

The specific activity of CA (i.e., activity of the free, immobilized onto the membrane, or the control filters, respectively) was assessed by monitoring the hydrolysis of p-NPA at 400 nm [47]. For the free enzyme, 0.1 mL of free CA was mixed with 0.8 mL Tris buffer and 0.1 mL of p-NPA (2.0 × 10^−2^ M in acetonitrile). For the immobilized enzyme, CA/MIL-160/Al_2_O_3_ membranes or CA/Al_2_O_3_ filters were immersed directly in mixture of 0.9 mL Tris buffer solution and 0.1 mL of p-NPA (2.0 × 10^−2^ M in acetonitrile) with varying concentrations of the substrate p-NPA (i.e., 1.0, 2.5, 5.0, 10, 20, and 40 mM, respectively). The absorbance was monitored every 2 min over a total period of 10 min. Additional blank experiments were conducted to estimate the self-dissociation of p-NPA in each of the solutions being used. Every experiment was repeated at least 3 times. 

### 2.7. Assessment of Enzyme-Based Conjugates

Chemical composition of the CA-based samples was evaluated by FTIR. For this, all of the CA-based samples were prepared by suspending the supports (Al_2_O_3_ and MIL-160/Al_2_O_3_) in CA-Tris (pH 7.5) solutions with the concentration of 0.1 mg/mL for 2 h under 100 rpm shaking conditions and at room temperature. After drying the samples at room conditions, they were tested using FTIR as described above. Controls of dried CA was also used. Scans were collected in the range of 400–4000 cm^−1^ at a resolution of 4 cm^−1^; a total of 128 scans were co-added to form the final spectrum of each sample. Experiments were performed in triplicates. AFM using Si tips were used to investigate the morphology of the samples containing immobilized enzyme, all in AC mode. Scans of 20 µm × 20 µm area or lower were acquired in air with subsequent order 3 image flattening. Height evaluation were performed using the Asylum software with subsequent MATLab analysis. 

### 2.8. Performance of the CA/MIL-160/Al_2_O_3_ Membrane or CA/Al_2_O_3_ Filters

Performance of as prepared CA/MIL-160/Al_2_O_3_ membrane or CA/Al_2_O_3_ filters used as controls for CO_2_ adsorption was evaluated using a modified method reported by Hou et al. [41]. Briefly, an in-house platform (Appendix A) composed of a membrane module, a gas chromatography unit (GC; Inficon, Micro GC Fusion, East Syracuse, NY, USA), mass flow meters, and pipelines was used. The membrane module was assembled from one right-angle flow rectangular manifold, three sets of stainless steel vacuum coupling O-ring (VCO) and stainless steel female nuts, and two stainless steel Swagelok tube fittings (Swagelok, Columbus, OH, USA). The GC unit was used for monitoring the CO_2_ concentration of the output gases, while the flow meters were used for controlling the content of the feeding gas. Herein, the feeding gas contained a mixture of 5% CO_2_ and 95% N_2_, with a controlled constant flow rate of 10 mL/min. This gas balance was chosen to mimic conditions in which the target molecule, i.e., CO_2_, corresponds to physical scenarios that are representative of current challenges in gas separation technologies [55,56], with carbon capture and storage (CCS) processes [55,57,58,59,60,61,62], for instance, targeting uptake of CO_2_ at dilute concentrations in power plants and chemical refineries [56,58,63,64,65]. The self-made membrane module also comprised a gas chamber for gas flow and liquid chamber for removing any product being released. The synthesized membrane was mounted in the self-built module with the CA/MIL-160 side in direct contact with the gas phase and the Al_2_O_3_ substrate side facing the liquid chamber. The overall and the specific CO_2_ adsorption rate were determined by the difference of the CO_2_ concentrations between the feeding and the output gas as shown in Equations (2) and (3), respectively:(2)$$x_{{co}_{2}}=\frac{C_{in,co_{2}}-C_{out,\;co_{2}}}{C_{in,\;co_{2}}}\times 100\%\;\;$$
(3)$$X_{{co}_{2}}=\frac{x_{co_{2}}}{m_{CA}}\times 100\%\;$$
where $C_{in,\;co_{2}}\;$ was the concentration of the feeding gas and $C_{out,\;co_{2}}$ was the concentration of the output gas, while $m_{CA}$ was the effective mass of enzyme used. Control experiments using free CA (0.1 mg/mL) were also performed. For this, 3 mL of 0.1 mg/mL CA solution was suspended into a glass reactor (Appendix A); the reactor was subsequently connected to the feeding gas and the rest of the configuration described above and used for evaluating the CO_2_ concentration changes. As above, the flow rate was maintained at 10 mL/min; experiments were performed in triplicates.

## 3. Results and Discussion

### 3.1. Synthesis and Characterization of the Components and the BioMembrane

We hypothesized that hydrophilic metal–organic Framework (MOF) MIL-160 can be used as a porous scaffold to increase functionality of carbonic anhydrase (CA) enzyme for its subsequent usage in carbon dioxide (CO_2_) gas adsorption. MIL-160s were selected based on their demonstrated capability to serve as flexible and hydrated “bridges” that ensure proton transfer to maintain enzyme functionality [46].

To demonstrate our hypothesis, we first synthesized the MIL-160/Al_2_O_3_ hybrids using a user-controlled hydrothermal method. Briefly, Al_2_O_3_ filters with uniform pore sizes (20 nm in diameter) were chosen as supports for MIL-160 “decorates”, with Al_2_O_3_ filters expected to allow coordination with 2,5-furandicarboxylic acid (FDCA), the known MIL-160 linker [46,66], through its Al metal ions [67,68]. It was expected that direct modification of the Al_2_O_3_ filter with FDCA would lead to controlled formation of MIL-160 onto the filters interface [69,70]. The hypothesis is supported by previous studies demonstrating that metal oxides can be used as metal sources for ZIF-8 formation [71], while ZrO_2_ can be used for UiO-66 [72] synthesis, as well as other reports [67,73] that showed the ability of porous Al_2_O_3_ filters to be modified with organic chemicals such as dopamine through a hydroxide-induced covalent reaction [67]. Furthermore, previous studies showed that a filter’s well-developed porosity and relatively high surface area, large pore volume, and abundant weak Lewis acid sites are essential to promote direct attachment of active molecules [74,75] (here, the FDCA). Such active molecules were further able to incorporate and disperse well into the individual filters’ channels to thus serve as active catalytic species [76]; such species are all needed for the MIL-160 formation.

Surface modification was confirmed by scanning electron (SEM) [77] and atomic force (AFM) [78] microscopy analysis. For the first, changes in Al_2_O_3_ thickness after FDCA functionalization were confirmed via cross-sectional SEM (Figure 1a), all relative to Al_2_O_3_ control (Figure 1b). For the second, morphology evaluation showed that FDCA functionalization led to changes in the 3D profiles of the Al_2_O_3_ filter surfaces (inserts, Figure 1a,b), as well as changes in the overall height profile distributions, with analysis revealing average height increases of about 4.7 times after filters modification with FDCA (Appendix A
Appendix A).

FDCA functionalization was further confirmed via Fourier-transform infrared spectroscopy (FTIR; Figure 1c), with analysis identifying characteristic peaks associated with the C=O at 1727 cm^−1^, C-O at 1228 cm^−1^, C=CH at 827 and 770 cm^−1^, C=C at 1577 and 1413 cm^−1^, and -OH at 3414 cm^−1^, respectively [79], all relative to the control FDCA (Appendix A
Appendix A). Peaks shifts were also visible and confirmed the coordination reaction. Indeed, previous analysis [46] showed that the peaks at 1674 and 1275 cm^−1^ associated with C=O and C−O bonds of FDCA disappeared in the spectra of MOFs, while the characteristic peaks at 1581 and 1414 cm^−1^ associated with the C=C bond of the furan ring and peaks at 962 and 825 cm^−1^ assigned to the C−H bond, respectively, were weakened relative to those of the FDCA used as control. Furthermore, energy-dispersive X-ray spectroscopy (EDS) analysis revealed concentration changes of the C, O, and Al elements for the functionalized FDCA/Al_2_O_3_ filters relative to the bare ones (Appendix A
Appendix A), with C and O increasing from 8.59 ± 0.36% to 12.45 ± 0.08% and 37.48 ± 0.19 % to 44.45 ± 0.11%, while the concentration of Al decreased from 53.93 ± 0.21% to 43.11 ± 0.09%, respectively.

FDCA/Al_2_O_3_ functionalized filter was subsequently used as support for the growth of MIL-160 under user-controlled hydrothermal method [46]. Formation of MIL-160 particles was confirmed by SEM analysis, with Figure 2a showing a representative image of the hybrids displaying clear edges and uniform distribution onto the filter. Control MIL-160 grown on Al_2_O_3_ filter (i.e., without any FDCA pretreatment) were edgeless and smaller in size and had poor uniformity (Figure 2a insert). The cross-section view of the hybrid indicated a continuous monolayer coverage with edge distribution resulting in an overall thickness changes of 13% to 15% relative to the thickness of the FDCA functionalized filter. Moreover, SEM analysis allowed particle size quantification, with the MIL-160 size in the hybrids being about 14.8 ± 0.9 µm while onto the filters, i.e., without FDCA pretreatment, being about 2.5 times smaller. AFM analysis also confirmed the edge distribution of the hybrids (Figure 2b). EDS mapping (Figure 2c) further demonstrated the formation of MIL-160/Al_2_O_3_ hybrids with layered distributions of C and O elements in the upper and Al in lower layer configuration, respectively. X-ray diffraction (XRD) confirmed the monophasic crystal nature of synthesized MOFs (Appendix A
Appendix A), with characteristic peaks recorded at 8.4° 9.4°, 11.98°, 15.2°, 18.8°, 22.8°, and 27°. Peaks were assigned to the planes (020), (011), (220), (031), (022), (051), and (502), respectively [80]. Lastly, the N_2_ adsorption analysis (Figure 2d) revealed Brunauer–Emmett–Teller (BET) and Langmuir surface area of MIL-160/Al_2_O_3_ hybrids of 214.58 and 264.70 m^2^/g, respectively, significantly larger than that of Al_2_O_3_ filters, which were 7.40 and 9.3 m^2^/g.

### 3.2. Functionality of the CA Enzyme at the BioMembrane Interface

Synthesized MIL-160/Al_2_O_3_ hybrids were subsequently used as supports for enzyme surface immobilization [46]. Immobilization was confirmed by FTIR and standard colorimetric assay. For the first, CA immobilized at the membrane interface showed broader peaks (1458–1735 cm^−1^) relative to the peaks of free enzyme, i.e., 1625 and 1523 cm^−1^ (Appendix A
Appendix A), presumably due to C=N stretching and N-H bending vibration [46,81] known to occur upon enzyme immobilization at interfaces. Additionally, control CA immobilized onto Al_2_O_3_ displayed narrower peaks relative to free enzyme in solution, most likely due to the enzyme’s deformation at such a filter interface [46]. The colorimetric assay confirmed enzyme loading at about 0.101 ± 0.006 mg per membrane, which was lower than that for the control filter that showed an average loading of 0.124 ± 0.006 mg, values recorded when both were offered 0.2 mg of enzyme during interfaces’ preparation. The higher enzyme loading observed for the CA/Al_2_O_3_ relative to CA/MIL-160/Al_2_O_3_ was presumable due to the porous structure of the Al_2_O_3_ filter; specifically, filter pore sizes of 25 nm could potentially allow entrapment of CAs (which are known to have individual diameters of ~4.5 nm [46]). Indeed, previous studies showed entrapment of enzymes in spaces as small as 3.0 and 4.1 nm when, for instance, microperoxidase-11 (MP-11) was interfaced with Tb-TATB [82], or in 9 and 15 nm in a periodic mesoporous organosilicas (PMOs) used for lipase immobilization [83], respectively. Considering that the porosity of MIL-160 is only 0.5 nm [84], such small pores are expected to provide additional space for gas adsorption analysis (as supported by BET evaluations) but, however, should not be prone for confinement of immobilized enzymes.

Activity evaluations (Figure 3a) and Michaelis–Menten kinetics of the CA/MIL-160/Al_2_O_3_ and CA/Al_2_O_3_ control were performed using the hydrolysis of para-nitrophenol acetate (*p*-NPA). Analysis showed that the enzyme immobilized at the membrane interface had a higher activity, higher *V*_max_ (5.66 ± 0.57 mM/mg enzyme/min.), and lower K_m_ (31.98 ± 4.46 mM) relative to the enzyme immobilized directly onto the filter (*V*_max_ of 4.16 ± 0.27 mM/mg enzyme/min and K_m_ of 39.97 ± 4.09 mM, respectively). The recorded surface-dependent activity and kinetic behavior was presumably due to enzyme deformation or enzyme entrapment (Figure 3b) at the two interfaces. In particular, the sharp edges and higher curvatures of MIL-160 framework could had led to a lower enzyme deformation and higher retained activity than the lower curvature of the Al_2_O_3_ used as control. This hypothesis is supported by previous work that showed that MIL-160 hydrophilicity does not only allow for a higher activity of immobilized enzymes, but further, their flexible and hydrated nature provide “bridges” to ensure proton transfer at their interfaces and lead to improved kinetic behavior [46]. Previous studies [85,86,87,88] also showed that supports with higher local curvatures were more beneficial for retaining enzyme activity, with Tadepalli et al. [85] demonstrating, for instance, higher activity for horseradish peroxidase absorbed on gold nanoparticles known to have higher surface curvature. Campbell et al. [87] also reported that the power density of an enzymatic biofuel cell could be 10-fold greater when glucose oxidase and bilirubin oxidase used to form enzymatically active anodes and cathodes, respectively, were immobilized onto graphene-coated single-wall carbon nanotube gels known to possess higher curvature than controls gold/multiwall carbon nanotube fiber paddles. In addition, our previous work [86] confirmed this principle by revealing a higher activity for soybean peroxidase immobilized onto carbon nanotube as opposite to the enzyme immobilized onto graphene oxide sheets. Lastly, complementary previous analysis showed that the functionality of the enzymes entrapped in porous matrices/supports was affected significantly by the substrate diffusion limitation to enzyme’s active site [89,90], with authors indicating that such diffusion limitations were due to the steric hindrances that the confined spaces imposed.

CO_2_ adsorption and reduction at the CA/MIL-160/Al_2_O_3_ membrane (i.e., changes in the CO_2_ concentration upon gas interaction with the user-formed membrane as well as changes in operational stability of the membrane upon extended gas exposure) were evaluated using an “in-house” platform (Appendix A). The CA/MIL-160/Al_2_O_3_ membrane was sealed in a tunnel-like chamber, with the CA-loaded side towards the feed gas and the other side being maintained in direct contact with stirred water. A membrane contactor was aimed to allow direct contact of the target gas (CO_2_) with the enzyme, while the MIL-160 porosity provided a hydrophilic microenvironment to retain immobilized enzyme’s activity. This lab-designed platform was thus hypothesized to directly assist with membrane hydration while also ensuring CO_2_ direct adsorption at the CA active site and eliminating the CO_2_ solvation [91] limitations. The mechanism of CO_2_ adsorption at the CA active site was previously described, with report showing that CO_2_ binds near the Zn ions [92] to lead to a Zn-coordinated transformation of bound CO_2_ into bicarbonate to be followed by the direct release of such product into the media (Appendix A) [25].

CO_2_ concentration changes at the membrane interface were evaluated by monitoring the concentration of CO_2_ in the output gas (i.e., after its exposure to the CA interface) through the gas chromatograph (GC); CO_2_ concentration in the feeding gas was maintained constant. The CO_2_ changes in concentration are shown in Figure 3c; specifically, analysis revealed about 5% change on average in the output CO_2_ concentration relative to the concentration used during membrane’s feeding. The concentration plateaued after about 54 ± 7.8 min of continuous membrane operation in the CO_2_ flow. In contrast, the changes at the CA-Al_2_O_3_ filter interface decreased drastically after only 50 min operation from an initial maximum of 1.4% to 0.5% and then subsequently plateaued after about 76.3 ± 3.2 min of operation. Controls Al_2_O_3_ and Al_2_O_3_/MIL-160 membranes showed no CO_2_ adsorption under the same conditions (Appendix A
Appendix A), thus eliminating concerns about possible absorbance of the gas at any of the two interfaces. Furthermore, free CA suspended in a lab-build bioreactor (Appendix A) at the same concentration as the immobilized CA showed a totally irregular CO_2_ change in concentration (Appendix A
Appendix A). Such patterns were presumably triggered by the Brownian motion and the molecular diffusion of the individual CA molecules suspended in water and their subsequent collision with CO_2_ at the gas-liquid phases.

Observed changes in the CO_2_ concentration at the CA/MIL-160/Al_2_O_3_ membrane interface relative to both free CA and CA/Al_2_O_3_ controls, respectively, were endowed by the design of the nanoscale hierarchical structure based on the MIL-160 and enzyme adsorption at such interfaces, respectively. Specifically, framework flexibility and intrinsic regular porosity not only provided a favorable microenvironment for both enzyme loading and functionality, but they also allowed for an active and viable transfer for both the reactant and the product to occur at the immobilized enzyme’s active site. These are supported by our previous study and our herein included BET and microscopy analysis showing that MIL-160 has high hydrophilicity and highly porous crystalline structure with defined square-shape sinusoidal channels of ~5 Å in diameter [84]. Moreover, water molecules played critical roles in maintaining CA activity since the framework, through its endowed hydrophilic and porous characteristics, served as a “bridge” for proton transfer for the enzyme regeneration step (Appendix A) to ensure CO_2_ hydration [46].

We also varied the concentration of CA offered during the immobilization process; analysis showed that even though enzyme loading increased from 0.115 to 0.403 mg/membrane when the CA amount in the feeding solution increased by 10 times for instance, the catalytic performance of the membrane did not change (Figure 3d). We thus hypothesize that what dictates efficiency is the enzyme packing/arrangement at the MOF interface. Indeed, the specific CO_2_ change in adsorption (Figure 3e) at the CA/MIL-160/Al_2_O_3_ membrane prepared using a CA concentration of 0.1 mg/mL reached the highest value of 1.26 mmol/mg CA/min. This was 1.6 and 5.0 times that of CA/MIL-160/Al_2_O_3_ membranes prepared under the CA concentrations of 0.5 and 1.0 mg/mL with effective enzyme loading being 0.042 ± 0.0003 and 0.117 ± 0.024 mg/membrane, respectively. Herein, the effective enzyme loading is referred to the amount of the accessible enzyme for CO_2_ adsorption as dictated by the enzyme–support specific interface. Such reduced enzyme efficiency was probably due to the intramolecular interaction of multilayered immobilized enzymes shown previously to cause activity losses at given interfaces [46,92]. Moreover, a larger enzyme loading could also result in mass transfer limitations due to the multilayer coverage to again lead to reduced overall efficiency of the immobilized enzymes [93,94].

Our results are superior to previous results from Hou et al. [95] or Wang et al. [96] (Appendix A
Appendix A), with the higher CO_2_ adsorption achieved by the design CA/MIL-160/Al_2_O_3_ membrane being presumably due to the advantages of the design itself which allows for both direct contact of CO_2_ with the enzyme as well as hydrophilic wetting through MIL-160. Specifically, Hou et al. [95] reported CO_2_ changes at a CA integrated Janus membrane (i.e., carbonic anhydrase (CA)-1H,1H,2H,2H-peruorododecyltrichlorosilane (FTCS)-carbon nanotubes (CNTs)-polyvinylidene fluoride (PVDF)) with authors studying the CO_2_ hydration performance when the enzyme was integrated into the hydrophobic polypropylene (PP) hollow fiber membranes. They showed a CO_2_ hydration rate of about 62% of the one achieved in this study while their effective membrane area was over 67 times larger than the one used herein [40]. Furthermore, Wang’s group achieved high efficiency of CO_2_ absorption flux of 2.5 × 10^−3^ mol·m^−2^·s^−1^, which is considerably reduced when compared to the one achieved in our study. Moreover, in their study the authors immobilized CA onto a polydopamine (PDA)/polyethylenimine (PEI)-polyvinylidene fluoride (PVDF) composite hollow fiber membrane through a crossing linking method [96], which provides limited scalable capabilities.

Our results demonstrated the potential of a user-designed membrane to be used as a bioreactor to provide a viable platform for efficient CO_2_ adsorption and reduction specifically considering that our previous analysis [46] that showed that thermal stability (investigated upon applying heat treatment to the conjugates, under various temperatures, for 30 min) is enhanced relative to the thermal stability of the free enzyme stored under the same conditions. In particular, we have previously shown [46] that CA-MIL-160 conjugates retained all of their original activity after heating treatment at 60 °C and only lost about 7% of their original activity after heating treatment at 70 °C. Our analysis demonstrated that the flexibility of the MOF interface protects the enzyme from rigid multiattachment and leads to cooperative enhancement of water affinity [97] with the excellent heat storage capacity of MIL-160 [80,84] further enhancing the high temperature tolerance of the immobilized CA relative to its free forms. Furthermore, our results indicate that such a bioreactor could possibly be used to evaluate CO_2_ hydration, with such process to be controlled by the functionality of the immobilized enzyme. Moreover, our study hints at the ability of the membrane to allow for evaluation of CO_2_ benign transformation, through direct measurement of bicarbonate product. Lastly, our analysis allows for complete integration and extension of the CO_2_ adsorption and transformation system through a parallel configuration of the membrane bioreactor design to thus allow for flow control and transformation in a modular geometry.

## 4. Conclusions

We developed an interface based on biocatalyst CA and a hydrophilic MOF, MIL-160, to be used for CO_2_ adsorption in synthetic environment and at room temperature. Our study successfully showed that membrane’s design and functionality depend on the materials characteristics, i.e., MIL-160 porosity and its ability to ensure a hydrophilic environment for the CA-driven reaction. We further demonstrated that enzyme’s efficiency is function of the geometry and morphology of the interface itself; we envision extension of such CA-based technology to industrial conditions upon adopting reactor design and modular implementation.

## Figures and Tables

**Figure 1 nanomaterials-11-01008-f001:**
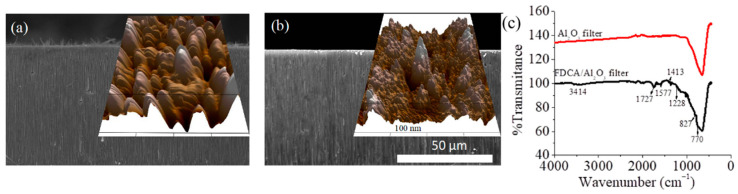
(**a**) SEM micrograph, cross-section of FDCA/Al_2_O_3_ functionalized filter; Insert: atomic force microscope (AFM) surface roughness; (**b**) SEM micrograph, cross-section of Al_2_O_3_ filter used as control; Insert: AFM surface roughness; (**c**) FTIR spectra of FDCA/Al_2_O_3_ functionalized filter relative to control.

**Figure 2 nanomaterials-11-01008-f002:**
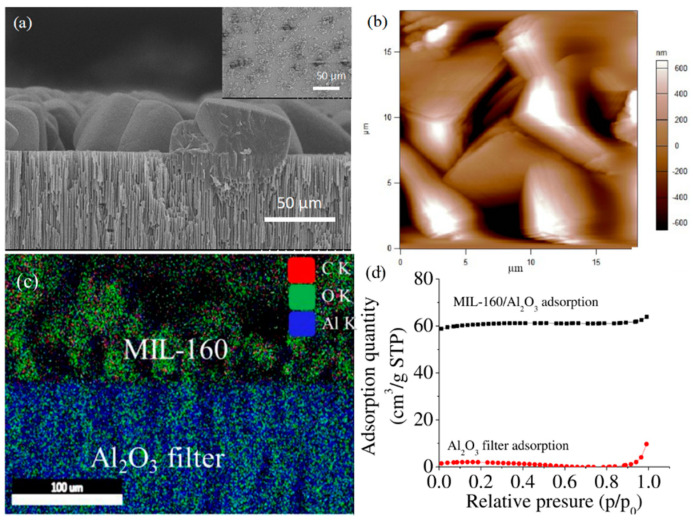
(**a**) Morphology of MIL-160/Al_2_O_3_ hybrid prepared upon FDCA surface modification of the Al_2_O_3_ filter; Insert: surface morphology of control MIL-160/Al_2_O_3_ hybrid prepared without FDCA surface modification; (**b**) control MIL-160/Al_2_O_3_ hybrid morphology with sheets-like conformations and sharp edges; (**c**) EDS mapping of the MIL-160/Al_2_O_3_ hybrid; (**d**) N_2_ adsorption isotherms of Al_2_O_3_ filter and MIL-160/Al_2_O_3_ hybrid.

**Figure 3 nanomaterials-11-01008-f003:**
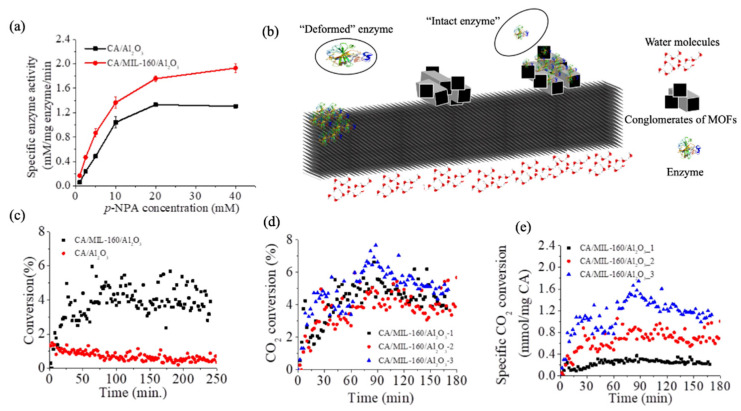
(**a**) Michaelis–Menten kinetics of CA/MIL-160/Al_2_O_3_ membrane relative to control CA/Al_2_O_3_ functionalized filter used as control; (**b**); proposed mechanism of binding and deformation of the CA at the membrane or filter interfaces, respectively; (**c**) CO_2_ concentration changes at the CA/MIL-160/Al_2_O_3_ and CA/Al_2_O_3_ interface and operational stability tested for 250 min; (**d**) CO_2_ changes in concentration at the CA/MIL-160/Al_2_O_3_ membranes interfaces (three tested systems prepared in the same conditions) and at different CA loadings; (**e**) specific CO_2_ adsorption efficiency (i.e., relative to the reported enzyme loading) for the CA/MIL-160/Al_2_O_3_ membranes.

## Data Availability

The data presented in this study are available on request from the first author.

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
