# Peer review of "Active Nanointerfaces Based on Enzyme Carbonic Anhydrase and Metal–Organic Framework for Carbon Dioxide Reduction"

_nanomaterials, 2021, doi:10.3390/nano11041008_

Round 1

Reviewer 1 Report

This manuscript reported the integration of carbonic anhydrases with a hydrophilic (MOF) for CO2 adsorption and reduction. The MOFs were found to create an active interface that improves enzyme properties. The authors are suggested to address the following comments.

The role of MOF is not clear to me. It appears that the majority of enzyme is loaded inside the alumina support. The MIL-160 has pores that are too small to be loaded with the enzyme.

The alumina membrane itself is hydrophilic. What is the reason or need to use MIL-160 to provide a hydrophilic microenvironment to retain immobilized enzyme’s activity?

What is the reason for using Al2O3 filter as the support, which is quite expensive compared to polymer filters?

What are the CO2 adsorption capacities and how are they compared with literature data for other CO2 adsorbents? Or what are the advantages of the enzyme compared to many other adsorbents? 

What is the reason for using 95% N2 and 5% CO2 mixed gas to evaluate the CO2 capture properties rather than using other compositions? Will the composition affect the specific CO2 adsorption rate and adsorption capacity?

XRD results are needed to confirm the formation of MOF phase.

What are the potential applications for this technology, CO2 capture from flue gas or from the air?

Is desorption necessary for such a system and how will it be performed?

Author Response

We thank the reviewer for the feedback on our work, the time allocated to improve the quality of our manuscript and for all the comments/ suggestions made. Please see attachment with specific comments as what changes were made. Again, thank you for your time.

Reviewer 2 Report

The manuscript by Liu et al. reports the preparation of carbonic anhydrase – MIL-160 composite material, which was characterized by a widely accepted for this type of materials set of methods - electronic and atomic microscopy, infrared spectroscopy and colorimetric analysis. The activity of the immobilized enzyme in CO2 conversion was evaluated.

I believe that the manuscript fits the scope of Nanomaterials, however, some comments may be added:

  1. The major advantage of MOF-immobilized enzymes over native enzymes is their enhanced chemical stability in a range of temperature and pH. The authors could comment or present some experimental results on pH and thermal stability of the carbonic anhydrase immobilized in MIL-160.
  2. Line 89 should read “2,5-furandicarboxylic acid”.
  3. In line 110 “of 55, 55 and 100 mM respectively” one of the concentrations (55 mM) should be different.

Author Response

We thank the reviewer for the feedback on our work, the time allocated to improve the quality of our manuscript and for all the comments/ suggestions made. Please see the attachment which explains how the comments were addressed; again, thank you for your time.

Reviewer 3 Report

This article reported by Liu et al describes a new strategy for developing enzyme-based interfaces as lab-built bioreactors capable to efficiently adsorb and reduce carbon dioxide at room temperature. The carbon dioxide reduction performances could be effectively improved by the integration of biocatalyst carbonic anhydrase (CA) and porous hydrophilic MOF, MIL-160, which could contribute to viable and functional surfaces for the adsorption and reduction of carbon dioxide due to both enzyme immobilization and interfaces functionality demonstration. Kinetics efficiency and system’s operational stability in user-set-up conditions were also studied. This work is of significant and suitable for publication in Nanomaterials. The following points should be concerned prior to publication.

--XRPD patterns for so-obtained MIL-160 should be provided.

--In Figure 3d and 3e, “CA/MIL-160/Al2O3-1”, “CA/MIL-160/Al2O3-2 and “CA/MIL-160/Al2O3-3” are not defined.

Author Response

We thank the reviewer for the feedback on our work, the time allocated to improve the quality of our manuscript and for all the comments/ suggestions made. Please see the attachment to evaluate how the comments were addressed; again, thank you for your time.

Round 2

Reviewer 1 Report

The authors have made the proper revisions.

Reviewer 2 Report

The authors have taken care to improve their manuscript and address all of the comments.